# ATTENTION SURGERY: AN EFFICIENT RECIPE TO LINEARIZE YOUR VIDEO DIFFUSION TRANSFORMER

## ABSTRACT

Transformer-based video diffusion models (VDMs) deliver state-of-the-art video generation quality but are constrained by the quadratic cost of self-attention, making long sequences and high resolutions computationally expensive. While linear attention offers sub-quadratic complexity, prior attempts fail to match the expressiveness of softmax attention without costly retraining. We introduce *Attention Surgery*, an efficient framework for *linearizing* or *hybridizing* attention in pretrained VDMs without training from scratch. Inspired by recent advances in language models, our method combines a novel hybrid attention mechanism—mixing softmax and linear tokens—with a lightweight distillation and fine-tuning pipeline requiring only a few GPU-days. Additionally, we incorporate a cost-aware block-rate strategy to balance expressiveness and efficiency across layers. Applied to Wan2.1 1.3B, a state-of-the-art DiT-based VDM, Attention Surgery achieves the first competitive sub-quadratic attention video diffusion models, reducing attention cost by up to 40% in terms of FLOPs, while maintaining generation quality as measured on the standard VBench and VBench-2.0 benchmarks.

## 1 INTRODUCTION

Video diffusion models (VDMs) have become a cornerstone of generative modeling, enabling high-fidelity, temporally coherent video synthesis for applications from entertainment to simulation. Early VDMs relied on U-Net backbones, but these architectures struggle to scale and capture long-range temporal dependencies. Recent advances favor Diffusion Transformers (DiTs) (Peebles & Xie, 2023). These models operate on spatiotemporal patches and provide global receptive fields from the outset. State-of-the-art systems such as *Wan2.1* (Wan et al., 2025), *CogVideoX* (Yang et al., 2025), *HunyuanVideo* (Lab, 2025), *PyramidalFlow* (Jin et al., 2025), and *Open-Sora Plan* (Lin et al., 2024) exemplify this trend, consistently outperforming U-Net-based models in quality and scalability. Recent surveys confirm this transition, highlighting DiTs as the dominant architecture for video generation (Wang et al., 2025b; Melnik et al., 2024).

While recent DiT-based video diffusion models deliver state-of-the-art quality, they come with substantial computational and memory costs that limit their practical applicability. A primary bottleneck lies in the self-attention mechanism, whose complexity scales quadratically with the sequence length, i.e., $O(N^2d)$ in time and $O(N^2)$ in memory, where $N$ denotes the number of tokens and $d$ the hidden dimension (Vaswani et al., 2017; Rabe & Staats, 2021). This issue is particularly severe in video diffusion, where the token count easily reaches tens of thousands due to the combination of spatial patches and multiple frames. Our profiling of large-scale DiT-based video diffusion models indicates that a vast proportion of compute is devoted to self-attention. For instance, in Wan2.1 1.3B, more than 76% of the total compute within the transformer blocks is attributable to self-attention alone. Even with optimizations such as FlashAttention (Dao et al., 2022), the quadratic scaling remains a fundamental barrier, constraining both training and inference when targeting higher resolutions, longer durations, or multi-shot videos. Consequently, reducing the attention cost without sacrificing model quality is critical for making video diffusion models more efficient and broadly deployable.

Although linear-time attention to address the quadratic cost of softmax attention (Katharopoulos et al., 2020; Choromanski et al., 2021) has been around for several years, no competitive video diffusion model based on linear attention exists today. Three factors explain this gap. **First**, training such

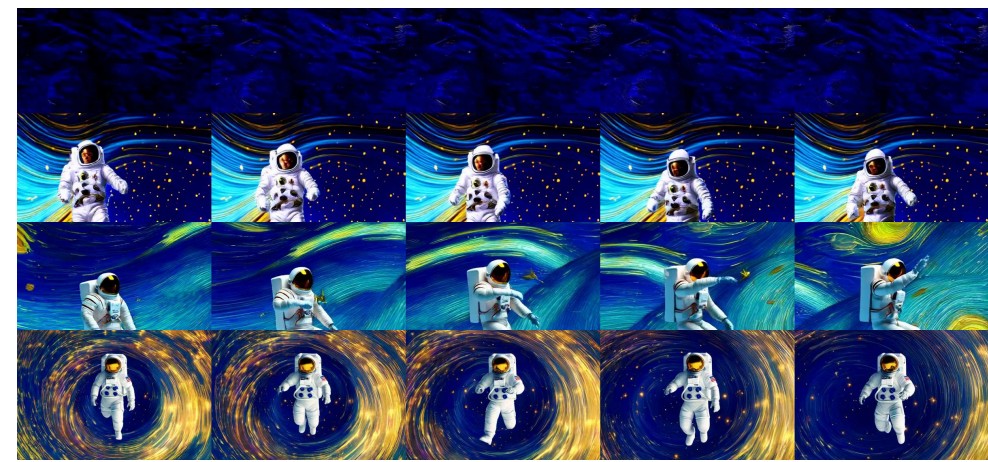

Figure 1: Impact of the proposed method components: attention distillation and hybrid attention. Prompt: *"An astronaut flying in space, Van Gogh style."*

models from scratch is prohibitively expensive: state-of-the-art video diffusion systems require hundreds of thousands to millions of GPU hours and massive curated datasets (Blattmann et al., 2023; Chen et al., 2024; Wan et al., 2025). **Second**, there is no practical method proposed for distilling softmax attention into linear attention under reasonable compute budgets for video diffusion. This difficulty arises because the exponential kernel underlying softmax requires an infinite-dimensional feature map for exact representation, making efficient approximations challenging (Han et al., 2024; Zhang et al., 2025). While linear attention in image diffusion (Li et al., 2023) and low-rank linearization in language models (Zhang et al., 2025) show promise, no analogous solution exists for the more complicated spatiotemporal token interactions in video diffusion. **Third**, lower expressiveness in linear attention often results in notable degradation in the attention transformation fidelity and consequently lower quality generations, specifically for videos with more complicated temporal signal dynamics.

To address these challenges, we propose an efficient attention surgery strategy – eliminating the need for extensive retraining from scratch – coupled with a novel efficient hybrid attention architecture inspired by recent developments in language modeling (Zhang et al., 2025). Intuitively, if a small subset of tokens retains full softmax attention while the rest use linear attention, the model can preserve global structure and fine-grained dependencies where needed, while scaling efficiently elsewhere. Our approach significantly narrows the quality gap between linearized and full softmax attention while achieving higher efficiency than the original softmax attention models. Importantly, it can be realized with modest compute –requiring less than 0.4k GPU hours for the overall surgery – making it practical for a wide range of research and industrial settings. We validate our method on *Wan2.1*, a state-of-the-art video diffusion model, demonstrating that our contributions are successfully applicable to transformer-based diffusion models. Figure 1 illustrates the obtained advantage.

Our main contributions are as follows:

- We introduce *attention surgery*, an efficient recipe that enables achieving competitive linear/hybrid models within only a few GPU days training on modestly-sized training datasets, liberalizing the process of such significant architectural operations.
- We propose a *novel hybrid attention* formulation with components carefully designed taking the intrinsics of videos into consideration.
- We propose a novel block-rate optimization strategy that adjusts the attention configuration of each block based on its transformation complexity, achieving the best accuracy–efficiency trade-off within a given compute budget.

## 2 RELATED WORK

**Efficient Attention.** Numerous approaches have been proposed to reduce the quadratic complexity of self-attention, for perception *e.g.* EfficientViT (Cai et al., 2023), PADRe (Letourneau et al., 2025), Performer (Choromanski et al., 2021), and Linformer (Wang et al., 2020), for image gen-

eration *e.g.* SANA (Li et al., 2023), LinGen Wang et al. (2025a), Grafting (Chandrasegaran et al., 2025), and for language modeling (Mercat et al., 2024; Wang et al., 2024; Yang et al., 2024; Zhang et al., 2024; 2025). Although these methods demonstrate the feasibility of sub-quadratic or more efficient attention, they typically require extensive training or training from scratch (e.g., SANA (Li et al., 2023)). In contrast, our work aims for lightweight distillation and fine-tuning of pre-trained softmax-attention-based models into an efficient hybrid attention design specifically tailored for video diffusion under modest compute budgets. Moreover, we target video generation task with unique challenges in modeling spatiotemporal token dependencies.

Linear recurrent models, such as SSM and RWKV, have recently emerged as efficient alternatives to self-attention, enabling the modeling of longer token sequences for high-resolution image generation (Fei et al., 2024b; Wang et al., 2024; Fei et al., 2024a; Zhu et al., 2025; Yao et al., 2025). However, due to the architectural differences between transformer blocks and SSM-based blocks (e.g., Mamba), distilling pre-trained DiT weights into these architectures typically requires extensive training. In contrast, our approach preserves the same underlying block structure, enabling effective distillation under a low-training regime. Furthermore, as highlighted in the seminal work on linear attention (Katharopoulos et al., 2020), there exists a strong connection between RNNs and causal linear attention, which allows a linearized causal attention mechanism to be deployed as an RNN during inference, a property that is particularly desirable for long video generation.

**Efficient Video Diffusion Models.** Recent large-scale video diffusion systems such as *CogVideoX* (Yang et al., 2025), *Open-Sora Plan* (Lin et al., 2024), *PyramidalFlow* (Jin et al., 2025), *LTX-video* (HaCohen et al., 2024), and *Wan2.1* (Wan et al., 2025) have advanced quality and scalability, but at enormous compute and memory cost. Mobile-oriented designs like *Mobile Video Diffusion* (Yahia et al., 2025), *MoViE* (Karjauv et al., 2024), *SnapGen-V* (Wu et al., 2025b;a), *AMD-HummingBird* (Isobe et al., 2025), and *On-device Sora* (Kim et al., 2025) explore lightweight architectures, yet remain non-DiT-based or rely on full quadratic attention, limiting scalability to long videos.

Prior work accelerates video generation via token merging (Bolya & Hoffman, 2023; Kahatapitiya et al., 2024; Ding et al., 2025), token downsampling (Crowson et al., 2024; Peruzzo et al., 2025), and attention tiling (Ding et al., 2025). *Efficient-vDiT* (Ding et al., 2025) improves DiT efficiency by tiling attention, achieving linear complexity and reducing memory overhead. In contrast, our hybrid attention with attention surgery remains quadratic in the worst case but reduces effective compute by applying softmax attention to only a fraction of tokens and linear attention to the rest. While not matching the asymptotic complexity of Efficient-vDiT, our method achieves better quality–efficiency trade-offs. Concurrent work M4V (Huang et al., 2025) accelerates video DiTs by distilling them into Mamba blocks. Despite our simpler block structure and lightweight training, we outperform M4V in both generation quality and efficiency.

## 3 METHODS

### 3.1 PRELIMINARIES: LINEAR ATTENTION

Let $x \in \mathbb{R}^{N \times F}$ represent a sequence of $N$ feature vectors of $F$ dimensional, and consider the $l$-th layer's transformer block, defined as:

$$T_l(x) = f_l\big(A_l(x) + x\big), \tag{1}$$

where $f_l(\cdot)$ applies a token-wise transformation, typically implemented as a small feedforward network, and $A_l(\cdot)$ denotes the self-attention operation, the only component that mixes information across the $N$ tokens, defined as:

$$A_l(x) = y = \text{softmax}\left(\frac{qk^\top}{\sqrt{D}}\right) v, \tag{2}$$

in which $q = xw_q$, $k = xw_k$ and $v = xw_v$ are linear projections of $x$ using learnable parameters $w_q, w_k \in \mathbb{R}^{F \times D}$, and $w_v \in \mathbb{R}^{F \times M}$. One can rewrite the softmax attention in the following form:

$$y_i = \frac{\sum_{j=1}^{N} \text{sim}(q_i, k_j)\, v_j}{\sum_{j=1}^{N} \text{sim}(q_i, k_j)}. \tag{3}$$

Following the kernel trick, reformulating $\text{sim}(q_i, k_j) = e^{q_i k_j^\top}$ reproducing the original softmax attention, to $\text{sim}(q_i, k_j) = \phi(q_i)\phi(k_j)^\top$ yields:

$$y_i = \frac{\phi(q_i)\sum_{j=1}^N \phi(k_j)^\top v_j}{\phi(q_i)\sum_{j=1}^N \phi(k_j)^\top}. \tag{4}$$

As observed, the terms $\sum_{j=1}^N \phi(k_j)^\top v_j$ and $\sum_{j=1}^N \phi(k_j)^\top$ are independent of the output index $i$ and thus can be precomputed and cached to achieve the above-defined reformulated attention in linear complexity. Note that $\phi(x)$ must be non-negative and the original linear attention paper by Katharopoulos et al. (2020) defines it as $\phi(x) = 1 + \text{elu}(x)$, however, the significant mismatch in expressiveness of the original similarity function $e^{q_i k_j^\top}$ and the elu-based $\phi(q_i)\phi(k_j)^\top$ results in substantial retraining compute and data requirement and/or inability to achieve the original quality observed from softmax attention.

## 3.2 HYBRID ATTENTION

**Attention Architecture**. To circumvent the aforementioned issues, and inspired by the recent developments in language models (Zhang et al., 2025), we define the hybrid attention by decoupling the full set of token indices $T = \{1 \dots N\}$ into softmax tokens $T_S$ and the rest as linear tokens, $T_L = T \setminus T_S$, as:

$$\hat{y}_i = \frac{\sum_{j \in T_S} \exp(q_i k_j^\top / \sqrt{D} - c_i)v_j + \phi_q(q_i)(\sum_{j \in T_L} \phi_k(k_j)^\top v_j)}{\sum_{j \in T_S} \exp(q_i k_j^\top / \sqrt{D} - c_i) + \phi_q(q_i)(\sum_{j \in T_L} \phi_k(k_j)^\top)}. \tag{5}$$

Here $c_i$ is the stabilizing constant computed as the maximum exponent. As opposed to Zhang et al. (2025), instead of defining $T_S$ as a local window around token $i$, we uniformly subsample tokens at *hybrid rate $R$*, as: $T_S = \{i \in T \mid i \mod R = 1\}$. This design ensures that higher-quality softmax tokens are distributed across the entire temporal span, providing global anchors that preserve motion coherence and prevent temporal drift — an issue that local windows often suffer from in video generation. Note that selecting subsampling rates $R$ is an important hyperparameter that indicates a trade-off between the fidelity of hybrid reconstruction of the original softmax and the corresponding computational efficiency. Higher values such as $R = 4$ or $8$ will ensure that the attention to only a small fraction of tokens scales quadratically that will notably decrease the compute burden. In contrast, a value of $R = 2$ can reconstruct the original softmax with higher accuracy, while still spending the quadratic terms on half of the tokens. Fig. 2 illustrates this.

**Characterization of $\phi$.** To enhance the expressiveness of linear attention, we define distinct learnable feature maps $\phi_q, \phi_k : \mathbb{R}^D \to \mathbb{R}^{P \cdot D'}$. Each map first applies a lightweight per-head embedding network (implemented as grouped $1 \times 1$ convolutions with non-linear activations) to produce an intermediate representation, which is then split into $P$ equal parts. Each part is raised to a different polynomial degree, from $1$ to $P$, and concatenated along the feature dimension. Formally, for an input $x \in \mathbb{R}^D$, we define:

$$\phi(x) = [(\psi_1(x))^1, (\psi_2(x))^2, \dots, (\psi_P(x))^P]^\top \in \mathbb{R}^{P \times D'},$$

where $\psi_i(\cdot)$ denotes the $i$-th learnable embedding slice produced by the shared embedding network. This polynomial expansion allows $\phi_q(q_i)\phi_k(k_j)^\top$ to approximate the large dynamic range of the exponential kernel $e^{q_i k_j^\top}$ more accurately than fixed ELU-based mappings.

## 3.3 ATTENTION SURGERY

To significantly decrease the required computational budget for training, we propose attention surgery as a framework that involves decoupling the process into three stages: *attention distillation*, *block-rate selection optimization* and *lightweight finetuning*.

**Attention Distillation**: Algorithm 1 details how we distill each softmax self-attention layer of a pretrained teacher diffusion model into the corresponding hybrid attention layer of the student. This step is crucial for maintaining the quality under aggressive attention linearization (e.g., large reduction factors or many transformed blocks), while keeping the training process lightweight. Note that the distillation of the student is done independently per isolated block, making it simple and scalable. Furthermore this stage of training only requires a set of prompts to train.

Figure 2: Overview of the attention distillation for the proposed attention surgery method. The example illustrates token separation with a hybridization rate of 3.

---

**Algorithm 1** Attention Distillation for isolated attention layer $l$ (Trainables $\phi = (\phi_q, \phi_k)$)

**Require:** Teacher params $\Theta^{\mathrm{T}}$; Student params $\Theta_l^{\mathrm{S}}$ of layer $l$ with frozen weights except for $\phi = (\phi_q, \phi_k)$; prompt distribution $\mathcal{P}$; noise distribution $\mathcal{N}$; sampling denoising steps set $\mathcal{T}$; batch size $m$; update repeats $U$; learning rate $\eta$;

1: **while** not converged **do**
2:      Randomly sample prompts $\{p^{(n)}\}_{n=1}^m \sim \mathcal{P}$ and initial noises $\{\varepsilon_0^{(n)}\}_{n=1}^m \sim \mathcal{N}$
3:      ▷ **Cache teacher trajectories:**
4:      $\{\{(x_t^{(n)}, y_{t,l}^{(n)})\}_{t\in\mathcal{T}}\}_{n=1}^m \leftarrow \text{TEACHERTRAJECTORY}(\Theta^{\mathrm{T}}, \{(p^{(n)}, \varepsilon_0^{(n)})\}_{n=1}^m, \mathcal{T})$
5:      **for** $u = 1$ **to** $U$ **do**          ▷ Multiple student updates from the cached batch
6:          $\hat{y}_{t,l}^{(n)} \leftarrow \text{STUDENTATTN}(\Theta_l^{\mathrm{S}}, x_t^{(n)})$          ▷ Using equation Eq. (5)
7:          $L \leftarrow \dfrac{1}{m\,|\mathcal{T}|} \sum_{n=1}^m \sum_{t\in\mathcal{T}} \left\| y_{t,l}^{(n)} - \hat{y}_{t,l}^{(n)} \right\|_1,$          ▷ Using Eq. (7) or Eq. (6)
8:          $\phi \leftarrow \phi - \eta \nabla_\phi L$
9:      **end for**
10: **end while**
11: **Return** $\phi = (\phi_q, \phi_k)$

---

As for the objectives, We define the *attention distillation* loss as:

$$\mathcal{L}_{\mathrm{ad}} = \log\left(1 + \left\| e^{q_i k_j^\top} - \phi_q(q_i)\phi_k(k_j)^\top \right\|_2^2\right), \tag{6}$$

where the logarithmic term mitigates numerical instabilities caused by large attention logits and gradients (Barron, 2025). However, given that matching the attention scores is a proxy optimization and the self-attention's weighted averaged hidden states are the target to match, one can alternatively define the *value distillation* objective as follows:

$$\mathcal{L}_{\mathrm{vd}} = \left\| y - \hat{y} \right\|_1, \tag{7}$$

where $y$ and $\hat{y}$ are defined according to Eqs. (2) and (5) respectively. Fig. 2 illustrates this. In practice, this distillation formulation significantly reduces the compute required for adapting large video diffusion models, enabling competitive hybrid attention efficiently. The most expensive variations of our attention surgery take less than 0.4k GPU hours.

**Heterogeneous Block-Rate Optimization.** Referring to Fig. 3, we observe that various blocks exhibit different reconstruction error values under different hybrid rates, each with their corresponding compute costs. How do we then decide upon the hybrid rates (8, 2, 4 or 1 (full softmax)) to form our hybrid architecture among the exponentially many combinations? We aim to optimize for an attention configuration for each transformer block under a global compute budget so as to minimize the approximated accumulated error throughout the network. More specifically, let the model have $B$ blocks indexed by $i \in \{1, \ldots, B\}$, and let $\mathcal{R}$ denote the set of candidate attention hybrid rates (e.g., $\mathcal{R} = \{1, 2, 4, 8\}$). For each block $i$ and rate $r \in \mathcal{R}$, we pre-compute:

- $c_{ir}$: estimated compute cost (e.g., FLOPs or latency),

- $e_{ir}$: estimated error relative to softmax, available from the attention distillation pretraining.

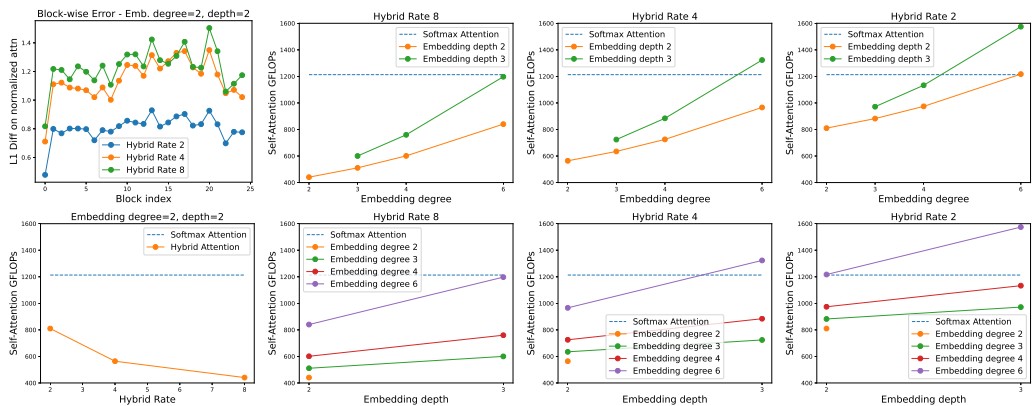

Figure 3: Per-block distillation error (top-left) and compute implications of $\phi$ architectural parameters and the attention hybrid rate.

Define binary decision variables $z_{ir} \in \{0, 1\}$, where $z_{ir} = 1$ iff block $i$ uses rate $r$. The optimization problem is:

$$\min_{\{z_{ir}\}} \sum_{i=1}^{B} \sum_{r \in \mathcal{R}} e_{ir} z_{ir} \qquad \text{s.t.} \sum_{i=1}^{B} \sum_{r \in \mathcal{R}} c_{ir} z_{ir} \leq \beta, \ \sum_{r \in \mathcal{R}} z_{ir} = 1 \ \forall i, \ z_{ir} \in \{0, 1\}. \tag{8}$$

This is a multiple-choice knapsack problem: select one rate per block to minimize the estimated accumulated error under a compute budget $\beta$, which can be efficiently solved (Kellerer et al., 2004). The solution to this optimization identifies the final configuration of the architecture with heterogeneous attention rates that provide the best accuracy/cost trade-off under the given budget. We call the block selection *homogeneous* if the set $\mathcal{R}$ consists of a single element, and *heterogeneous* otherwise.

**Lightweight Fine-tuning**. After the pretraining distillation stage and the block-rate selection optimization, we shape the final Hybrid DiT architecture with hybrid attention modules that are distilled. However, while the pretraining distillation helps the overall model to keep the general structure of the scenes, the details will be far from perfect, as the layers are pretrained in isolation. Now fine-tuning the whole DiT architecture on a modest set of prompt/video pairs for only a few hundred iterations will recover the lost generation quality.

## 4 EXPERIMENTAL SETUP

**Evaluation.** We evaluate our proposed attention surgery and hybrid attention methods using the Wan2.1 1.3B video diffusion model (Wan et al., 2025). For state-of-the-art comparisons, we generate videos at the original Wan resolution and length ($81 \times 480 \times 832$) using the full set of extended prompts from the VBench and VBench-2.0 benchmarks (Huang et al., 2024; Zheng et al., 2025).

In addition to quantitative evaluation, we conduct a blind user study to assess visual quality and prompt alignment. We randomly select 50 prompts from VBench and present participants with paired videos, asking them to choose their preferred video or indicate no significant difference. In total, we collect 562 paired comparisons.

To enable large-scale ablation studies, we fine-tune a lower-resolution model producing $320 \times 480$ frames on a subset of VBench comprising one-fifth of the original prompts. To mitigate performance fluctuations from short training runs, we evaluate each configuration at four iteration counts—i.e., 400, 600, 800, and 1,000—and report averaged results.

**Datasets.** For fine-tuning low-resolution models, we use a 350K subset of the video dataset from Open-Sora Plan (Lin et al., 2024). For high-resolution fine-tuning, we use 22K synthetic video samples generated by Wan2.1 14B, with prompts drawn from the same source as the low-resolution dataset.

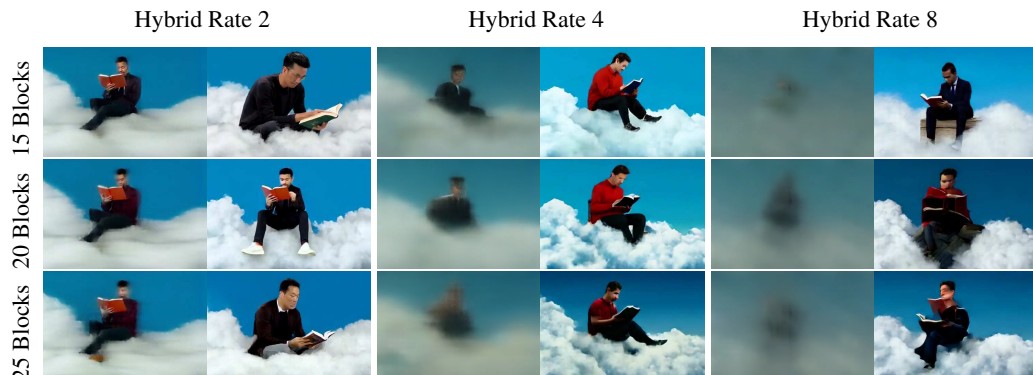

Figure 4: Sample qualitative video frames from hybrid models with varying numbers of hybrid blocks (15, 20, 25) and hybrid rates (2, 4, 8). For each configuration, the left frame shows the result after layer-wise attention distillation, and the right frame shows the result after 1,000 fine-tuning iterations. Prompt: *A man is reading a book sitting on the cloud*.

| Models with 2B–5B parameters | Total↑ | Quality↑ | Semantic↑ |
|---|---|---|---|
| Open-Sora Plan V1.3 | 77.23 | 80.14 | 65.62 |
| CogVideoX 5B | 81.91 | 83.05 | 77.33 |
| CogVideoX1.5 5B | 82.01 | 82.72 | 79.17 |
| **Models up to 2B parameters** | | | |
| Efficient VDiT | 76.14 | – | – |
| Open-Sora V1.2 | 79.76 | 81.35 | 73.39 |
| LTX-Video | 80.00 | 82.30 | 70.79 |
| SnapGenV | 81.14 | 83.47 | 71.84 |
| Hummingbird 16frame | 81.35 | 83.73 | 71.84 |
| Wu et al. – Mobile | 81.45 | 83.12 | 74.76 |
| CogVideoX 2B | 81.55 | 82.48 | 77.81 |
| PyramidalFlow | 81.72 | 84.74 | 69.62 |
| M4V | 81.91 | 83.36 | 76.10 |
| Wan2.1 1.3B | 83.31 | 85.23 | 75.65 |
| Wan2.1 1.3B* | 82.47 | 83.33 | 79.01 |
| + Attention Surgery (15×R2) | 82.40 | 83.43 | 78.28 |

Table 1: Comparisons with SOTA efficient video diffusion models. All metrics are extracted from reported numbers, except for 'Wan2.1*', which is our reproduction using the same evaluation pipeline and parameters as our variations.

| Prompt Dimension | Preference % | | |
|---|---|---|---|
| | Ours | No preference | Wan2.1 |
| Appearance style | 51.8 | 19.6 | 28.6 |
| Color | 52.2 | 21.7 | 26.1 |
| Human action | 16.2 | 54.1 | 29.7 |
| Object class | 30.3 | 30.3 | 39.4 |
| Overall consistency | 30.5 | 45.8 | 23.7 |
| Scene | 10.0 | 40.0 | 50.0 |
| Spatial relationship | 43.9 | 33.3 | 22.8 |
| Subject consistency | 35.7 | 21.4 | 42.9 |
| Temporal flickering | 20.7 | 56.0 | 23.3 |
| Temporal style | 28.3 | 39.1 | 32.6 |
| Total | 31.0 | 39.7 | 29.3 |

Table 2: Results of the method-blinded human visual preference study over 562 paired comparisons. Rows correspond to subsets filtered by different VBench prompt dimensions.

**Model Hyperparameters.** We experiment with converting different numbers of transformer blocks to hybrid attention: 15, 20, and 25 out of the 30 blocks in Wan2.1 1.3B. For hybrid blocks, we explore hybridization rates of 2, 4, and 8. Based on empirical analysis of the impact of $\phi_k$ and $\phi_q$ transformation complexity on generation quality, we find that a lightweight 2-layer MLP with degree-2 polynomial features is sufficient, adding approximately 2.4M parameters per converted block. Unless otherwise stated, we use value distillation loss during pretraining. Additional hyperparameter details are provided in Appendix A.

## 5 RESULTS

**Distillation vs Finetuning: Qualitative**. Fig. 4 shows the qualitative results of our models with different hybrid rates and number of hybrid blocks. It can be noted that in most cases, attention distillation alone is insufficient, leaving a noticeable gap that can be resolved by the following lightweight finetuning. Furthermore, lower-rate hybrid attention, e.g. $R = 2$, provides a much better reconstruction right after distillation, but the quality gap narrows significantly by the lightweight finetuning.

**Comparison with SOTA**. Tab. 1 shows a comparison of one of the variations of our model with the state-of-the-art methods on VBench. Results indicate that our hybrid operated models are compet-

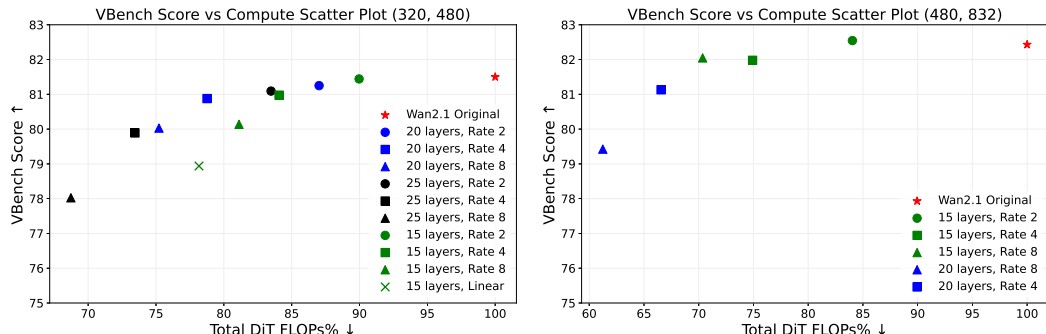

Figure 5: The total DiT FLOPs percentages versus the VBench score of original Wan2.1 1.3B model compared to various hybrid configurations or 320×480 (left) and 480×832 (right) resolutions.

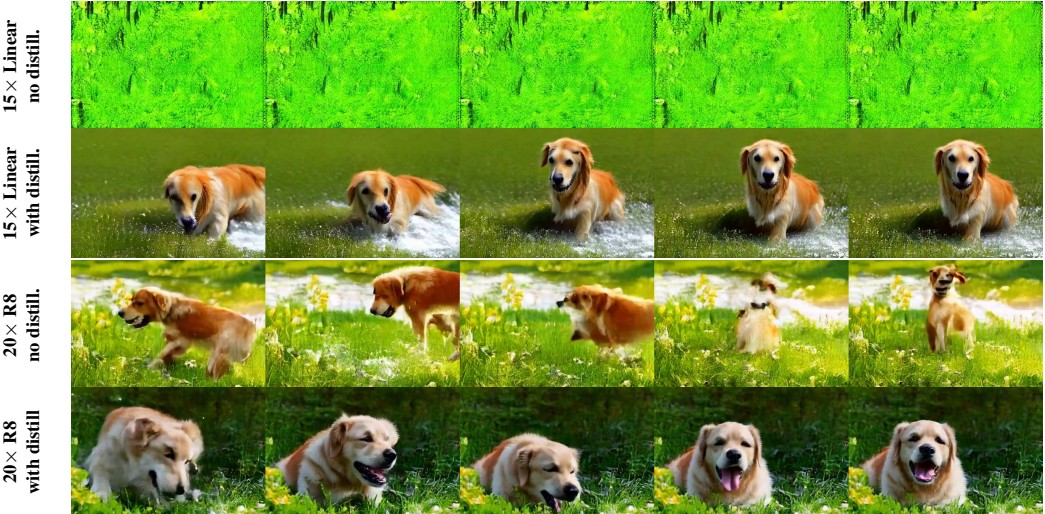

Figure 6: Qualitative illustration of impact of attention distillation on two hybrid architecture instances (15×Linear and 20×R8). Prompt: *"A playful golden retriever bounds through a sunlit meadow, its fur gleaming in the warm afternoon light."*

itive with state-of-the-art efficient video diffusion models, while one of the variations (15×R2) is equivalent to the original Wan2.1* model it's based on. On VBench-2.0 15×R2 gets total score of 55.1, comparable to the original 56.0. Please refer to Supplementary for full set of evaluations.

**User Study** The user study comparing the original Wan2.1 and our 15×R2 hybrid model, presented in Tab. 2, also reveals no overall preference for the original model.

**Various Hybrid Architectures**. Fig. 5 illustrates the compute-quality trade-off for various hybrid attention configurations, and compares them against the original baseline Wan2.1 model with the softmax attention blocks. As it appears, one can replace the original attentions with hybrid attention for half of the blocks or more while the quality of the generated videos is not significantly impacted. This is obtained with less than 0.4k GPU hours in contrast to estimated hundreds of thousands to millions of GPU hours to train SOTA text-to-video diffusion models from scratch. An extensive set of qualitative videos is available in the appendix and supplementary materials.

**Impact of Attention Distillation**. In Tab. 3, we show the impact of attention distillation with two different setups: 15 blocks with learnable linear attention (15× Linear), and computational equivalent of 20 blocks of hybrid attention with rate 8 (20×R8). As it can be noted, not performing distillation significantly impacts the quality of the outputs. A qualitative example showing multiple video frames from each of the 4 variations is illustrated in Fig. 6.

**Impact of Block Rate Optimization**. To assess the effectiveness of the heterogeneous block-rate selection strategy proposed in Sec. 3.3, we used 4 different budget setups, as reported in Tab. 4.

| Converted Blocks | Attention | Distillation | Total↑ | Quality↑ | Semantic↑ |
|---|---|---|---|---|---|
| 15 | Linear | × | 59.7 | 69.7 | 20.0 |
| 15 | Linear | ✓ | **78.9** | **82.2** | **65.9** |
| 20 | Hybrid $R = 8$ | × | 77.3 | 80.2 | 65.9 |
| 20 | Hybrid $R = 8$ | ✓ | **80.0** | **81.7** | **73.2** |

Table 3: VBench scores comparison of linear/hybrid models with and without attention distillation.

| | Block selection | Total↑ | Quality↑ | Semantic↑ | | Block selection | Total↑ | Quality↑ | Semantic↑ |
|---|---|---|---|---|---|---|---|---|---|
| $15 \times R4$ | homogeneous | 81.0 | 82.4 | 75.1 | $20 \times R8$ | homogeneous | 80.0 | 81.7 | 73.2 |
| | **heterogeneous** | **81.9** | **83.2** | **76.4** | | **heterogeneous** | **80.9** | **82.3** | **75.4** |
| $20 \times R4$ | homogeneous | 80.9 | 81.9 | **76.61** | $25 \times R4$ | homogeneous | 79.9 | 81.1 | **75.1** |
| | **heterogeneous** | **81.1** | **82.5** | 75.2 | | **heterogeneous** | **80.2** | **82.1** | 72.5 |

Table 4: Impact of the proposed heterogeneous block-rate selection strategy under different budget constraints. Our method consistently leads to marginally better total VBench score.

We compare our optimization-based method with a simpler homogeneous baseline: conversion of blocks with the lowest error after attention distillation under given hybrid rate, similar to Fig. 3. The proposed method consistently (albeit incrementally) improves VBench total scores.

| Converted Blocks | Hybrid Rate | Distillation Loss | Total ↑ | Quality ↑ | Semantic ↑ | Dynamic Degree ↑ |
|---|---|---|---|---|---|---|
| 20 | R8 | Attention distil. | 79.5 | 81.5 | 71.5 | 37.5 |
| 20 | R8 | Value distil. | **80.0** | **81.7** | **73.2** | **66.1** |
| 15 | R8 | Attention distil. | **81.2** | **82.8** | **74.6** | 51.8 |
| 15 | R8 | Value distil. | 80.1 | 81.9 | 72.9 | **66.1** |

Table 5: Comparison of distillation loss types, as measured by VBench scores.

| $\phi$ Specification | | VBench Total↑ | | | Parameters ↓ | FLOPs ↓ |
|---|---|---|---|---|---|---|
| Poly. degree | MLP layers | 10×R8 | 15×R8 | 20×R8 | (M) | (G) |
| 6 | 3 | 82.1 | 80.8 | **80.3** | 15.4 | 387 |
| 4 | 2 | **82.3** | 81.6 | **80.3** | 3.9 | 70 |
| 3 | 2 | **82.3** | **81.9** | 79.8 | 2.4 | 60 |
| 2 | 2 | 82.1 | 81.5 | 80.2 | **1.2** | **30** |

Table 6: VBench total scores for various hybrid architectures with different complexities of the learnable $\phi$ transformation, varying in MLP depth and polynomial degree.

**Attention Distillation Loss**. To evaluate the impact of the loss function choice, Tab. 5 exposes experiments on two different hybrid architectures (15×R4 and 20×R4). Despite the total VBench score differs marginally, we observe that using value distillation loss consistently results in videos with significantly more motion. Furthermore, our qualitative observations show a significantly larger number of sampled videos with attention distillation have cartoonish style, which does not necessarily hurt VBench scores but leads us to prefer using the value distillation loss variant.

$\phi$ **Transformation Characterization**. The complexity of the $\phi$ transformation function can have notable impact on the expressiveness of the linear/hybrid attention as well as the corresponding compute cost, as shown in Fig. 3. As it is observable in Tab. 6, it appears that a 2-layer MLP with polynomial degree of 2 constitutes a competitive variation while being the most efficient.

# 6 CONCLUSION AND FUTURE WORK

In this work, we introduced *attention surgery*, a framework for efficiently replacing self-attention—the most computationally expensive component in transformer blocks—with linear or hybrid counterparts, while requiring only moderate compute and data. We demonstrated that this approach enables efficient variants of video diffusion models that achieve performance competitive with state-of-the-art baselines on the widely used VBench benchmark. Looking ahead, combining linearity with causality in attention mechanisms could enable RNN-like video diffusion models whose attention cost does not grow with video length. Exploring causal formulations of linear and hybrid attention is therefore an exciting direction for future research.

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

# A APPENDIX

## A.1 TRAINING DETAILS AND HYPERPARAMETERS

Except for the ablation studies we characterize $\phi$ with a 2-layer MLP and a polynomial degree of 2, and make two separate transformations for keys and queries ($\phi_k$ and $\phi_q$) per hybrid block.

**Pretraining distillation stage**. We train each block independently and all the parameters are frozen except for the $\phi_k$ and $\phi_q$, for which we use the AdamW optimizer Loshchilov & Hutter (2019), batch size of 1 and a learning rate of 1e-3, with the value distillation objective, as detailed in equation 7 to train. To extract teacher activations for distillation, we sample using 50 denoising steps with a guidance scale of 5, employing the Euler Ancestral Discrete Scheduler to integrate the reverse diffusion process.

**Finetuning**. Within the finetuning process, we finetune all parameters of the hybrid DiT, including the $\phi$'s, the feed-forward MLP, etc., with a batch size of 16, AdamW optimizer and a learning rate of 1e-5 and bf16 mixed precision training. The model is trained for only 1000 iterations.

## A.2 QUALITATIVE SAMPLES

Figures 8 to 25 show uniformly spaced frames from videos generated by the original Wan2.1 1.3B and different variations of our hybrid attention models (15×R2, 15×R4, 15×R8, 20×R4, and 20×R8), for 18 different prompts on the original 480×832 resolution. All the videos corresponding the demonstrated frames, are available as video files in the attached supplementary materials.

## A.3 DETAILED VBENCH COMPARISON

Figure 7 shows a selected subset of our hybrid models compared to the original Wan2.1 1.3B model, on each of the comparison dimensions. The experiment is with the full VBench set and at the original 480×832 resolution.

## A.4 DETAILED VBENCH-2.0 COMPARISON

Tabs. 7 to 9 demonstrate fine-grained results on the recent VBench-2.0 benchmark at original resolution of 480×832. We generated videos with the original Wan2.1 1.3B model and our 15×R2 modification using the same sampler hyperparameters. We observe that our hybrid model experiences an insignificant drop in performance as measured by Total score.

## A.5 USE OF LARGE LANGUAGE MODELS

We used Microsoft Copilot (a large language model) to aid in polishing the writing of this submission. The model was employed solely for improving clarity and readability; all ideas, technical content, and conclusions are our own.

| Method | Human Identity | Dynamic Spatial Relationship | Complex Landscape | Instance Preservation | Multi-View Consistency | Human Clothes | Dynamic Attribute | Complex Plot |
|---|---|---|---|---|---|---|---|---|
| Wan2.1 1.3B* | 63.5 | 25.1 | 16.4 | 86.0 | 9.6 | 97.9 | 49.1 | 11.3 |
| Attention Surgery (15×R2) | 62.7 | 25.1 | 18.4 | 84.8 | 7.1 | 97.1 | 44.0 | 13.2 |

Table 7: VBench-2.0 results (part 1/3).

| Method | Mechanics | Human Anatomy | Composition | Human Interaction | Motion Rationality | Material | Diversity | Motion Order Understanding |
|---|---|---|---|---|---|---|---|---|
| Wan2.1 1.3B* | 72.4 | 80.6 | 48.4 | 71.7 | 40.8 | 69.4 | 49.1 | 32.0 |
| Attention Surgery (15×R2) | 66.4 | 77.0 | 46.4 | 70.3 | 41.4 | 67.3 | 48.5 | 33.7 |

Table 8: VBench-2.0 results (part 2/3).

| Method | Camera Motion | Thermotics | Creativity Score | Commonsense Score | Controllability Score | Human Fidelity Score | Physics Score | Total Score |
|---|---|---|---|---|---|---|---|---|
| Wan2.1 1.3B* | 32.1 | 61.7 | 48.7 | 63.4 | 34.0 | 80.7 | 53.3 | 56.0 |
| Attention Surgery (15×R2) | 29.0 | 70.5 | 47.5 | 63.1 | 33.4 | 79.0 | 52.8 | 55.1 |

Table 9: VBench-2.0 results (part 3/3).

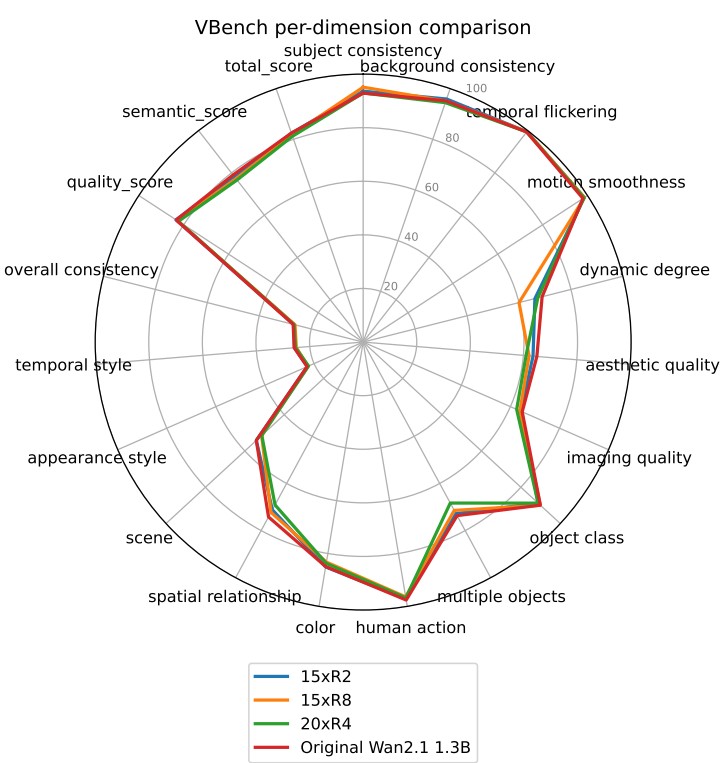

Figure 7: Radar plot comparing a subset of our hybrid models with the original Wan 1.3B model on the full VBench set and 480×832 resolution

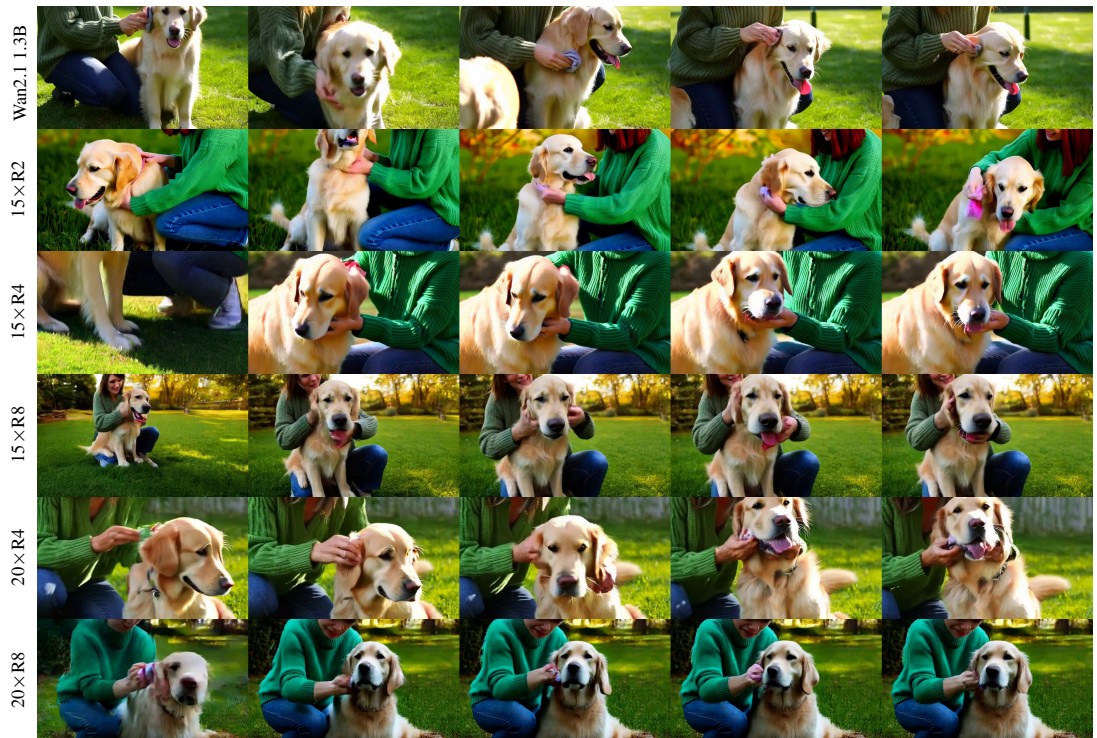

Figure 8: Qualitative videos comparing original Wan2.1 1.3B model to our various hybrid variations for input prompt *A person is grooming dog*

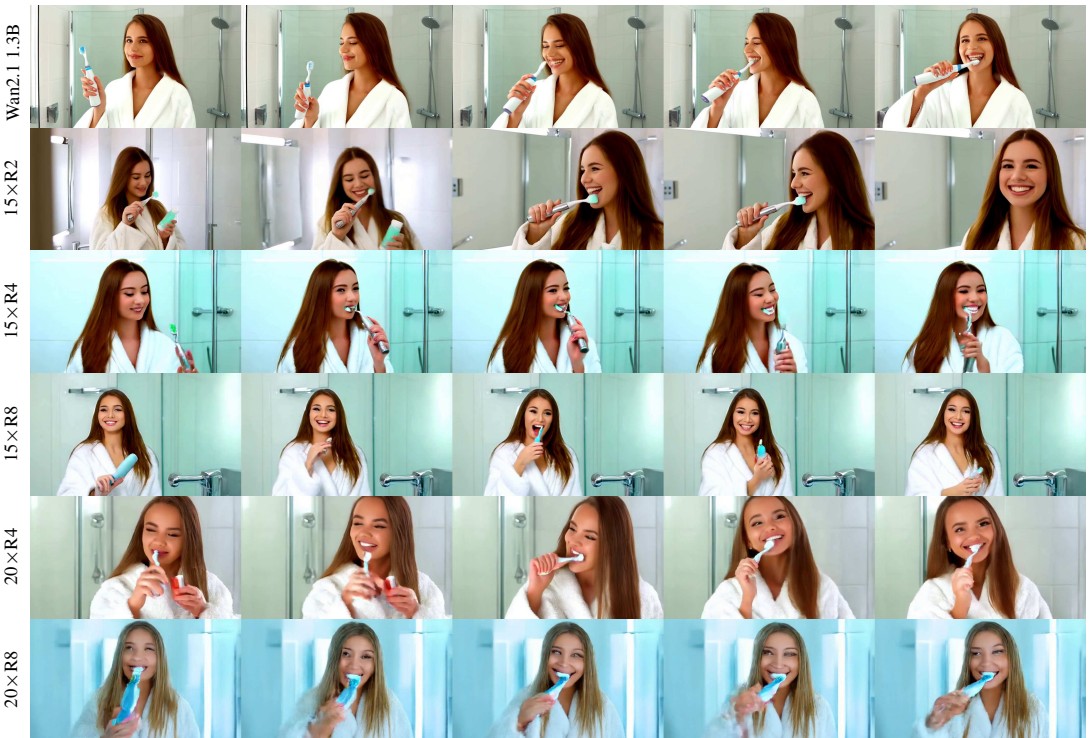

Figure 9: Qualitative videos comparing original Wan2.1 1.3B model to our various hybrid variations for input prompt *a person and a toothbrush*

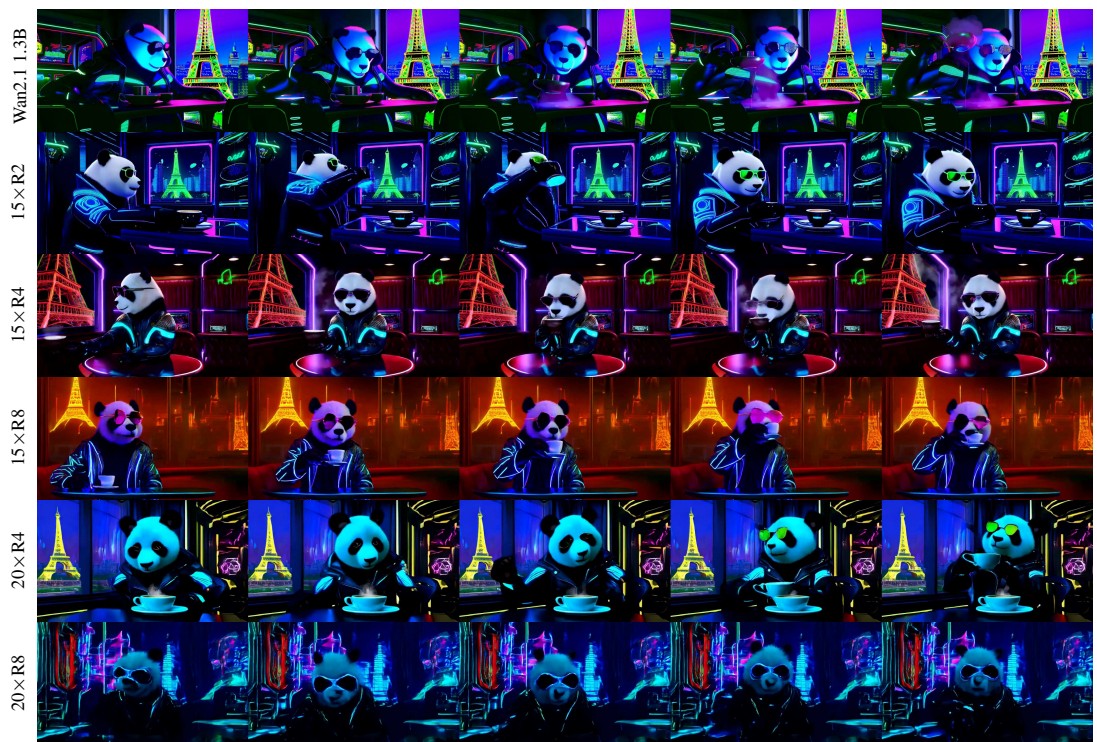

Figure 10: Qualitative videos comparing original Wan2.1 1.3B model to our various hybrid variations for input prompt *A panda drinking coffee in a cafe in Paris, in cyberpunk style*

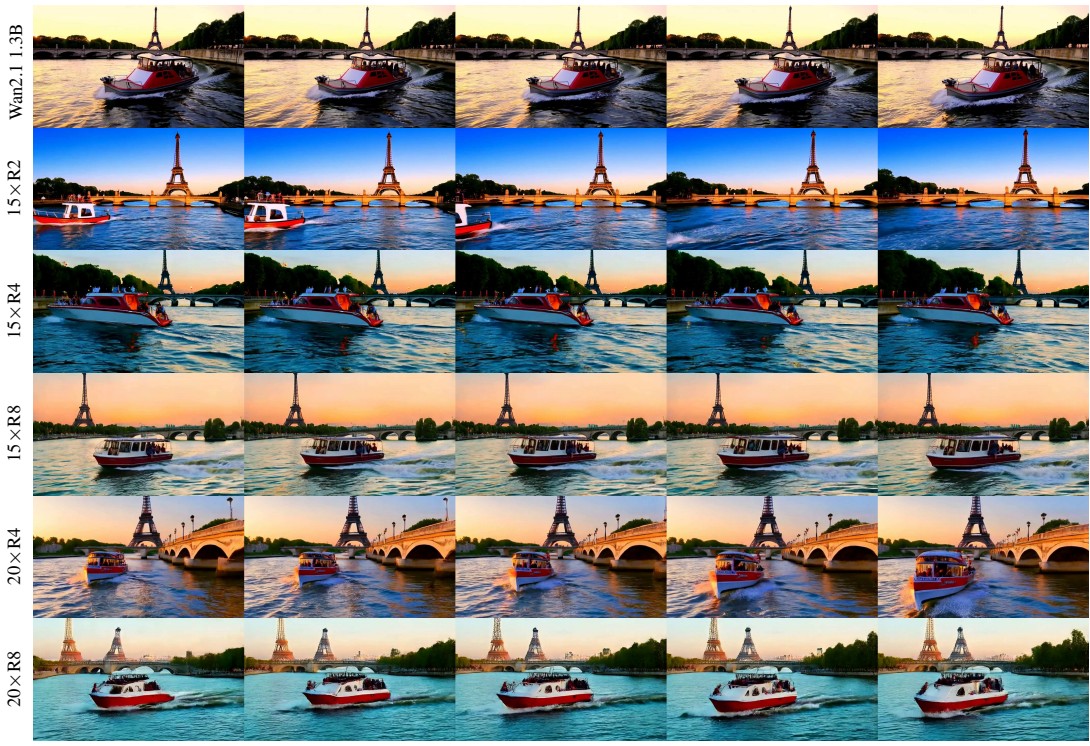

Figure 11: Qualitative videos comparing original Wan2.1 1.3B model to our various hybrid variations for input prompt *A boat sailing leisurely along the Seine River with the Eiffel Tower in background*

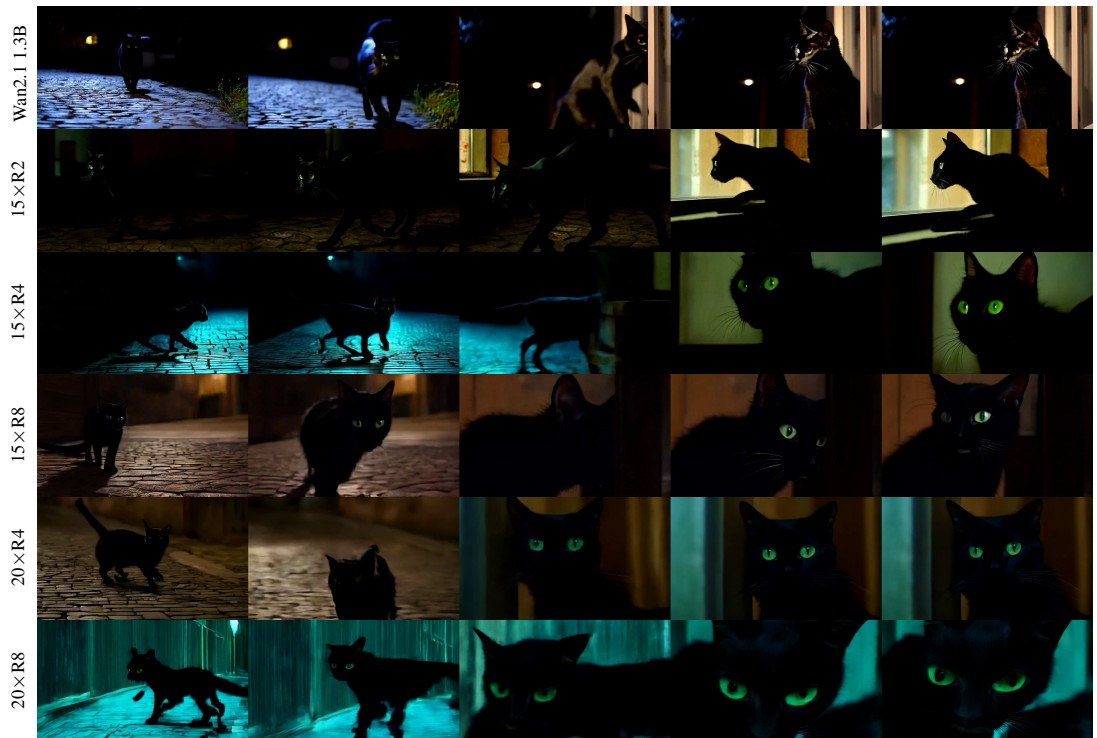

Figure 12: Qualitative videos comparing original Wan2.1 1.3B model to our various hybrid variations for input prompt *a black cat*

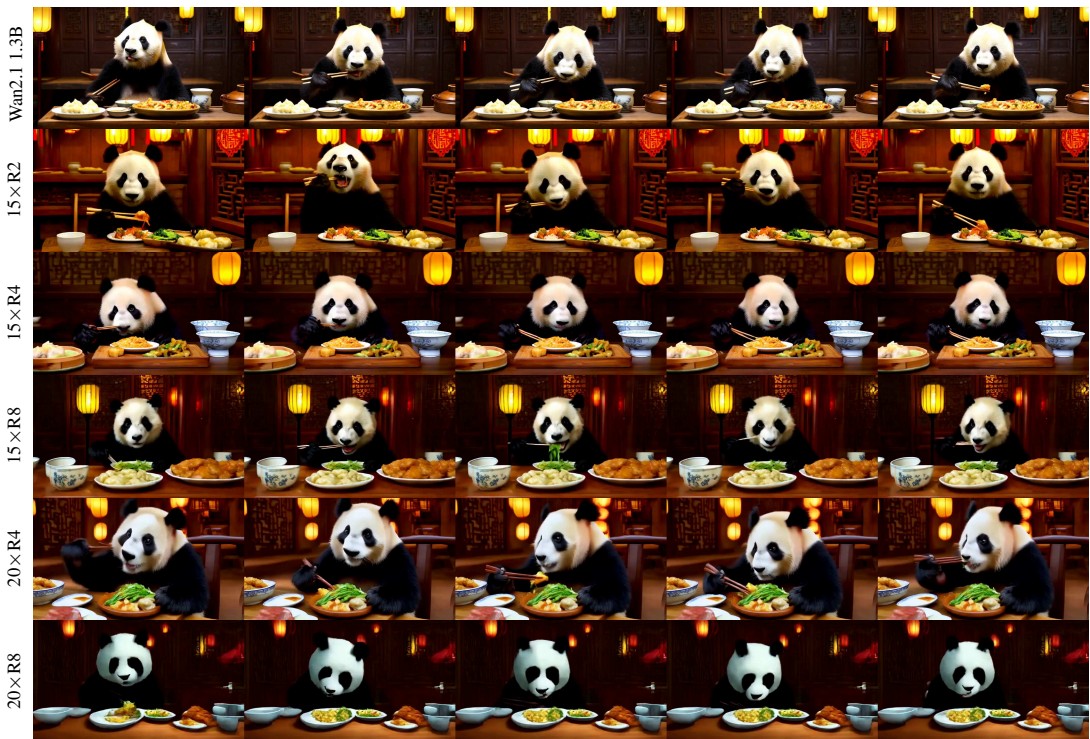

Figure 13: Qualitative videos comparing original Wan2.1 1.3B model to our various hybrid variations for input prompt *A cute fluffy panda eating Chinese food in a restaurant*

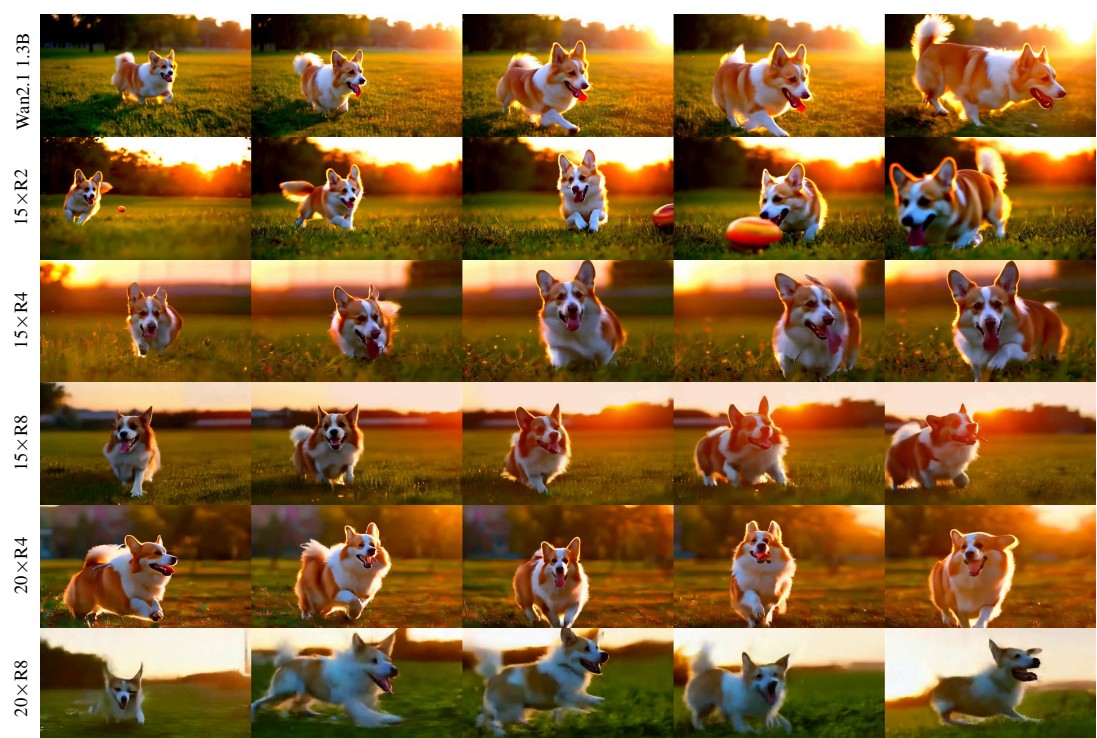

Figure 14: Qualitative videos comparing original Wan2.1 1.3B model to our various hybrid variations for input prompt *A cute happy Corgi playing in park, sunset, with an intense shaking effect*

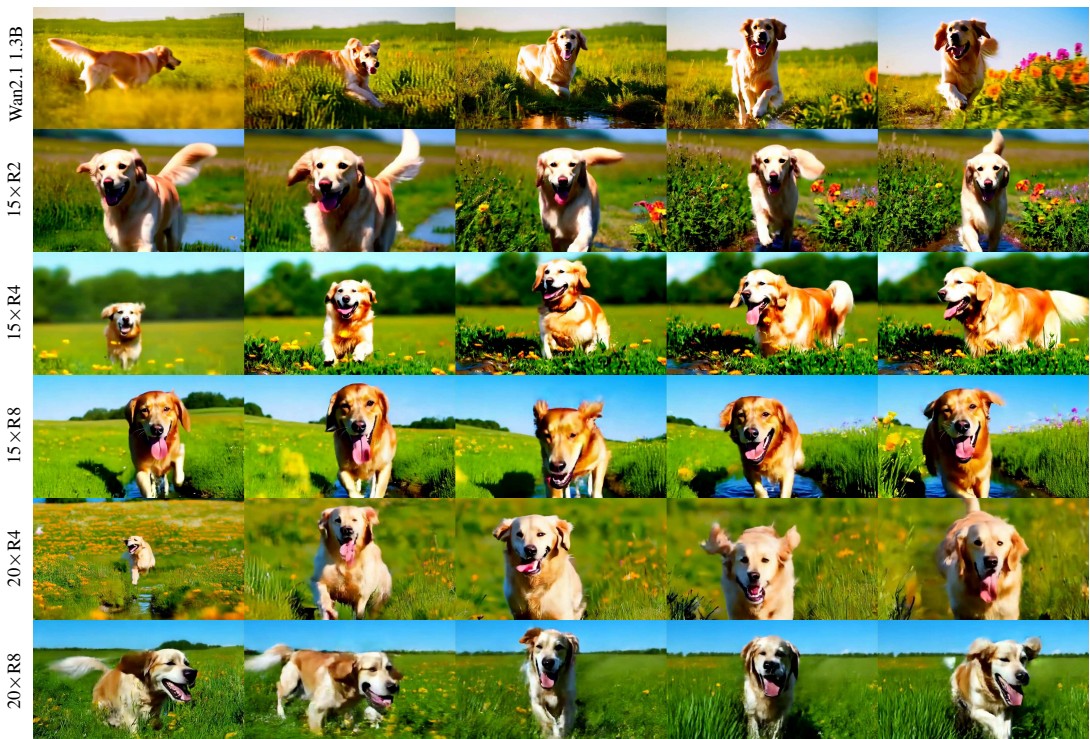

Figure 15: Qualitative videos comparing original Wan2.1 1.3B model to our various hybrid variations for input prompt *a dog running happily*

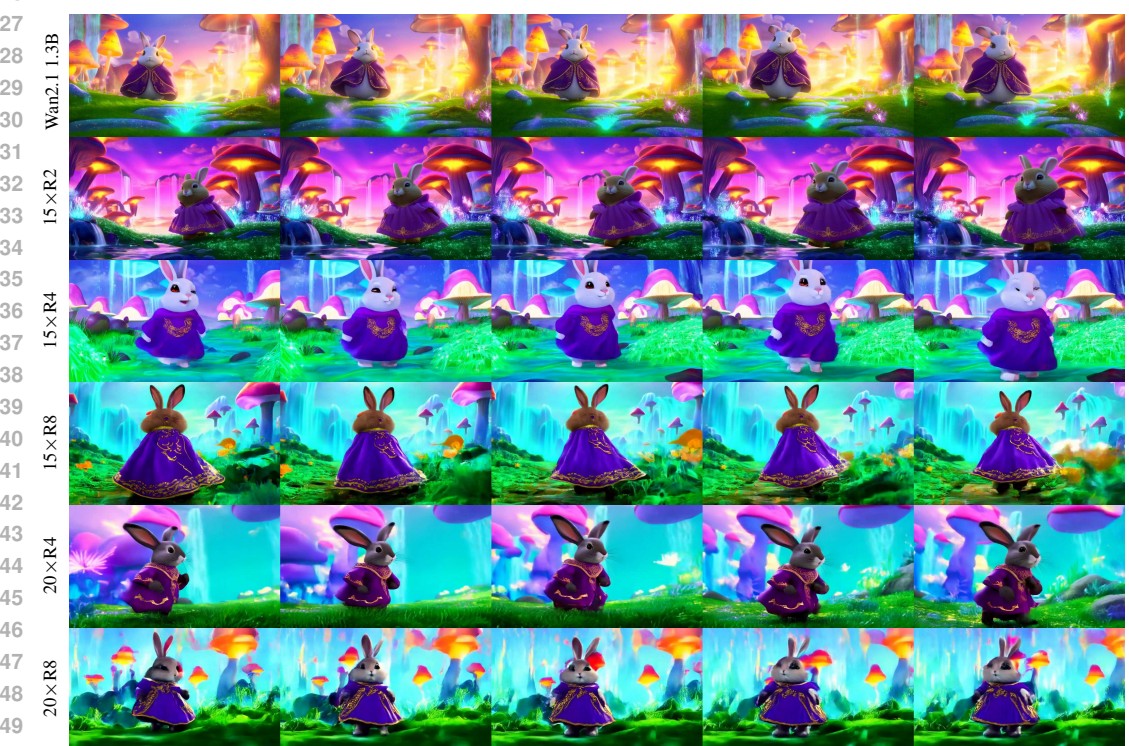

Figure 16: Qualitative videos comparing original Wan2.1 1.3B model to our various hybrid variations for input prompt *A fat rabbit wearing a purple robe walking through a fantasy landscape.*

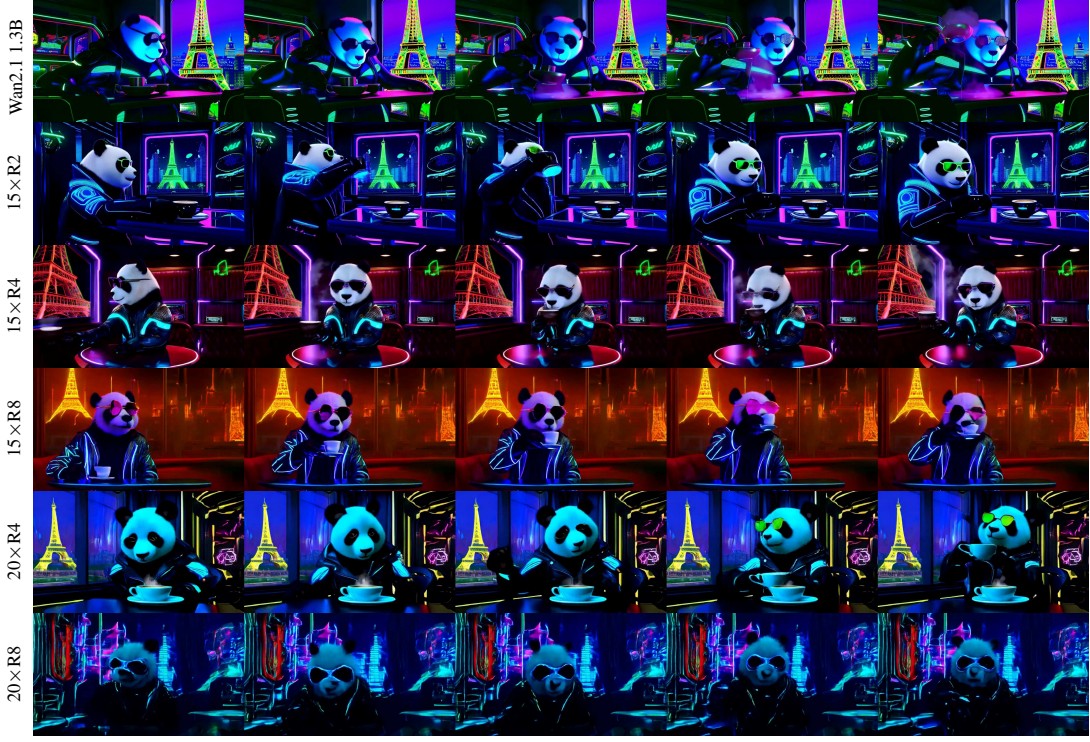

Figure 17: Qualitative videos comparing original Wan2.1 1.3B model to our various hybrid variations for input prompt *A panda drinking coffee in a cafe in Paris, in cyberpunk style*

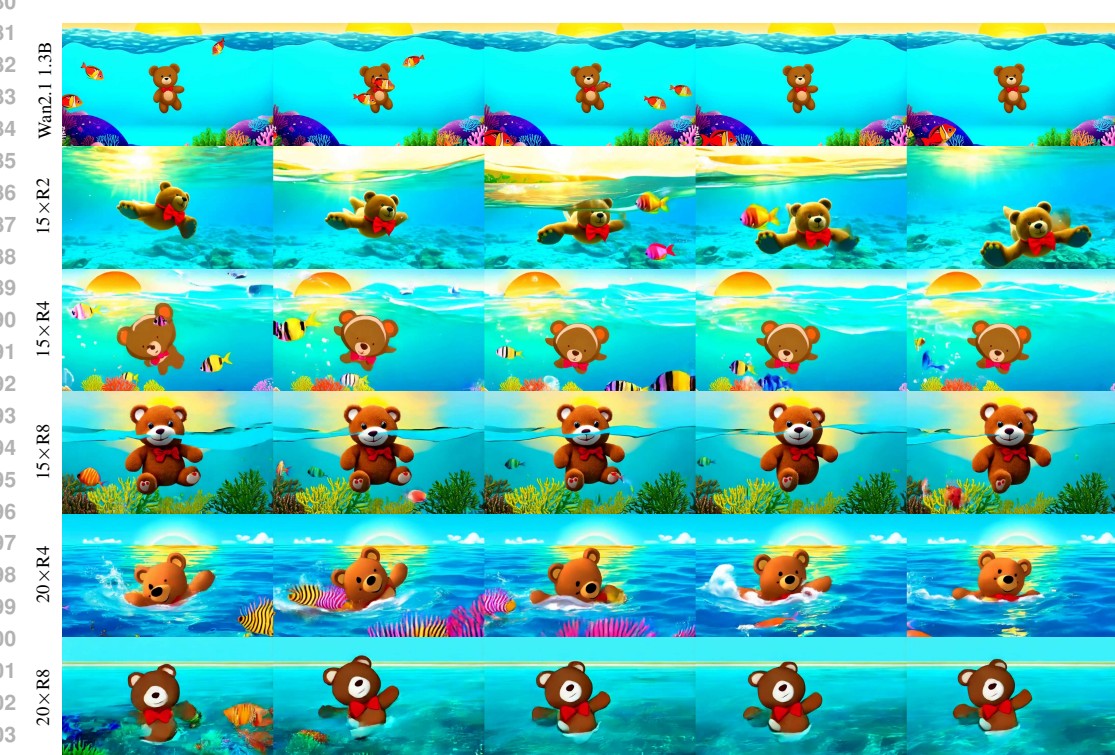

Figure 18: Qualitative videos comparing original Wan2.1 1.3B model to our various hybrid variations for input prompt *a teddy bear is swimming in the ocean*

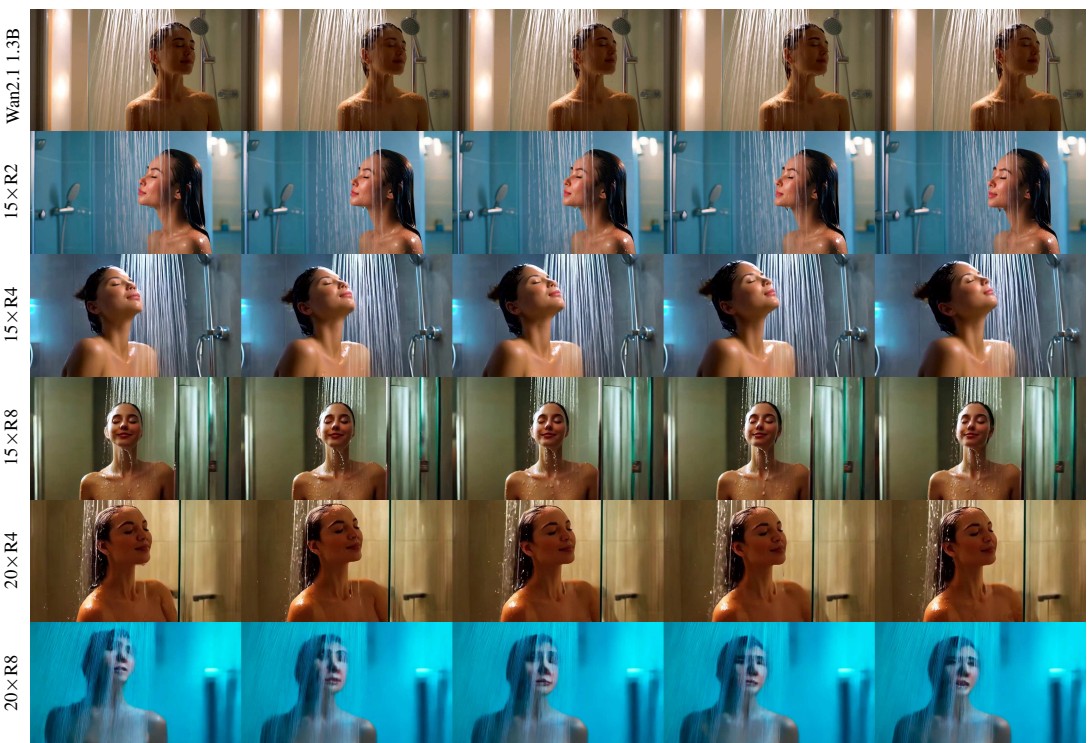

Figure 19: Qualitative videos comparing original Wan2.1 1.3B model to our various hybrid variations for input prompt *A person is taking a shower*

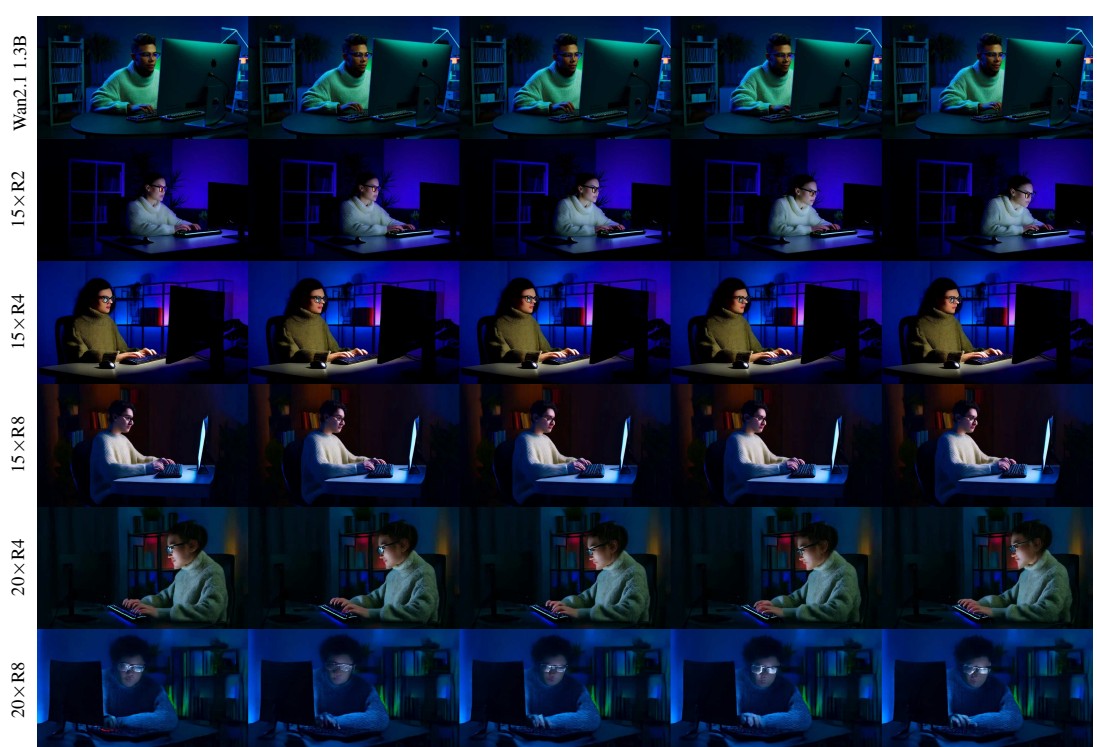

Figure 20: Qualitative videos comparing original Wan2.1 1.3B model to our various hybrid variations for input prompt *A person is using computer*

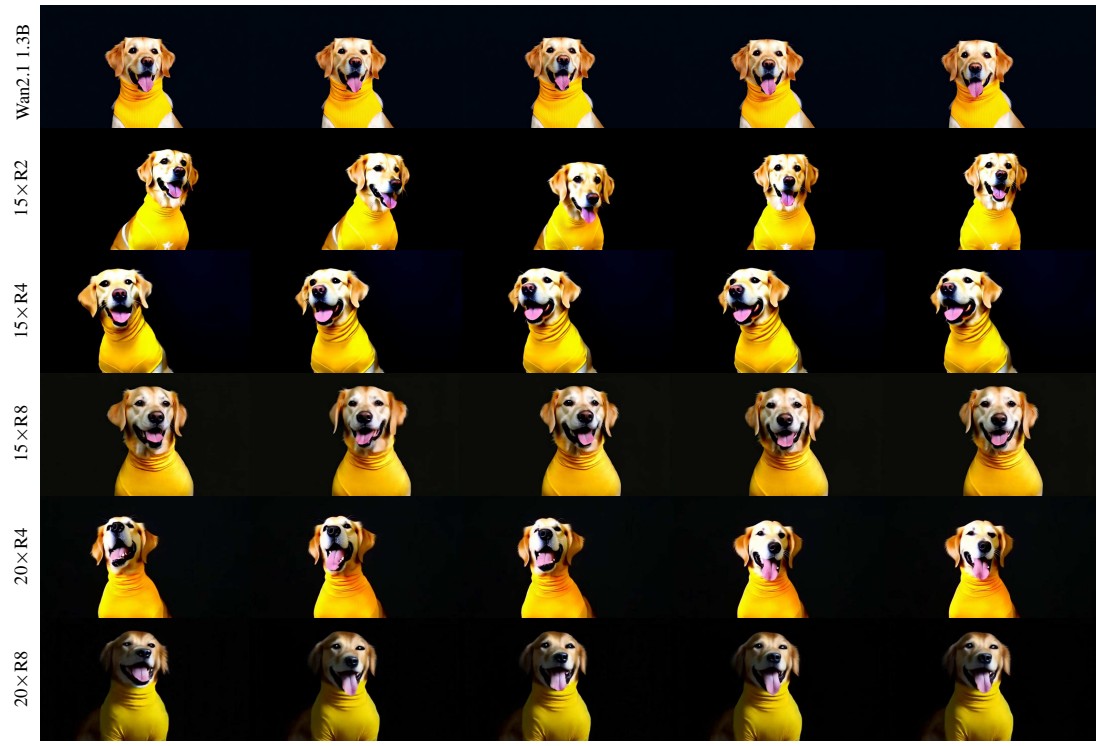

Figure 21: Qualitative videos comparing original Wan2.1 1.3B model to our various hybrid variations for input prompt *happy dog wearing a yellow turtleneck, studio, portrait, facing camera, dark background*

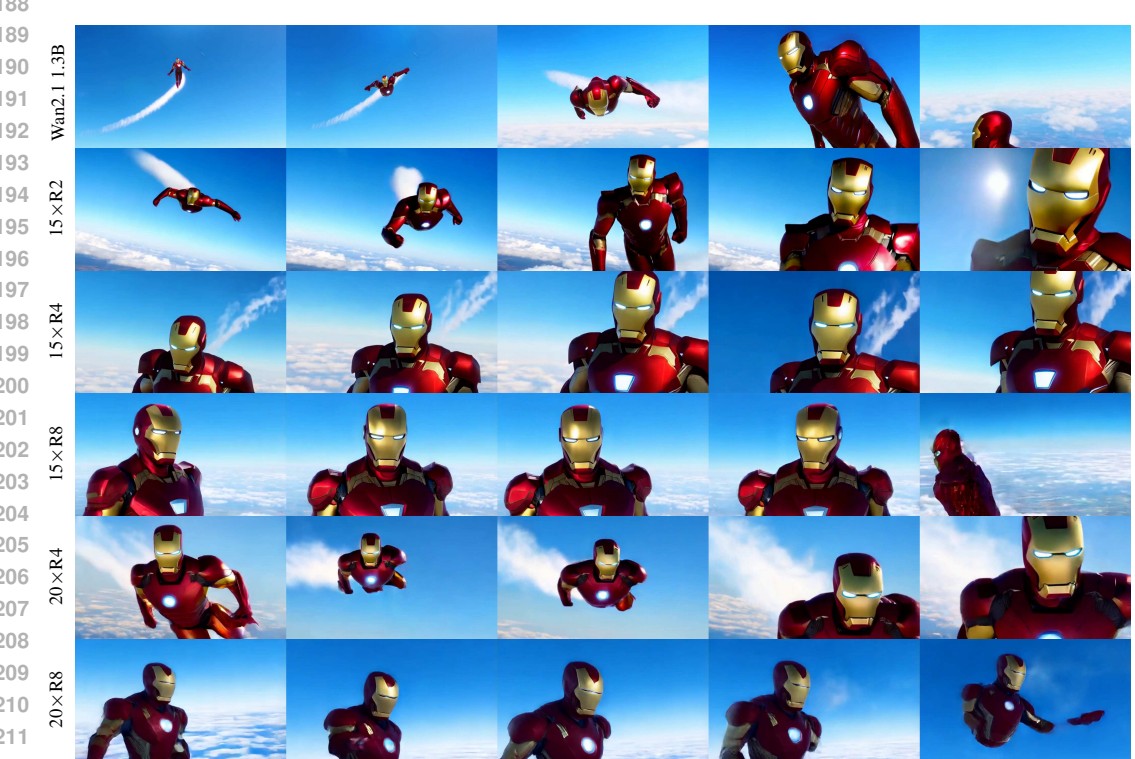

Figure 22: Qualitative videos comparing original Wan2.1 1.3B model to our various hybrid variations for input prompt *Iron Man flying in the sky*

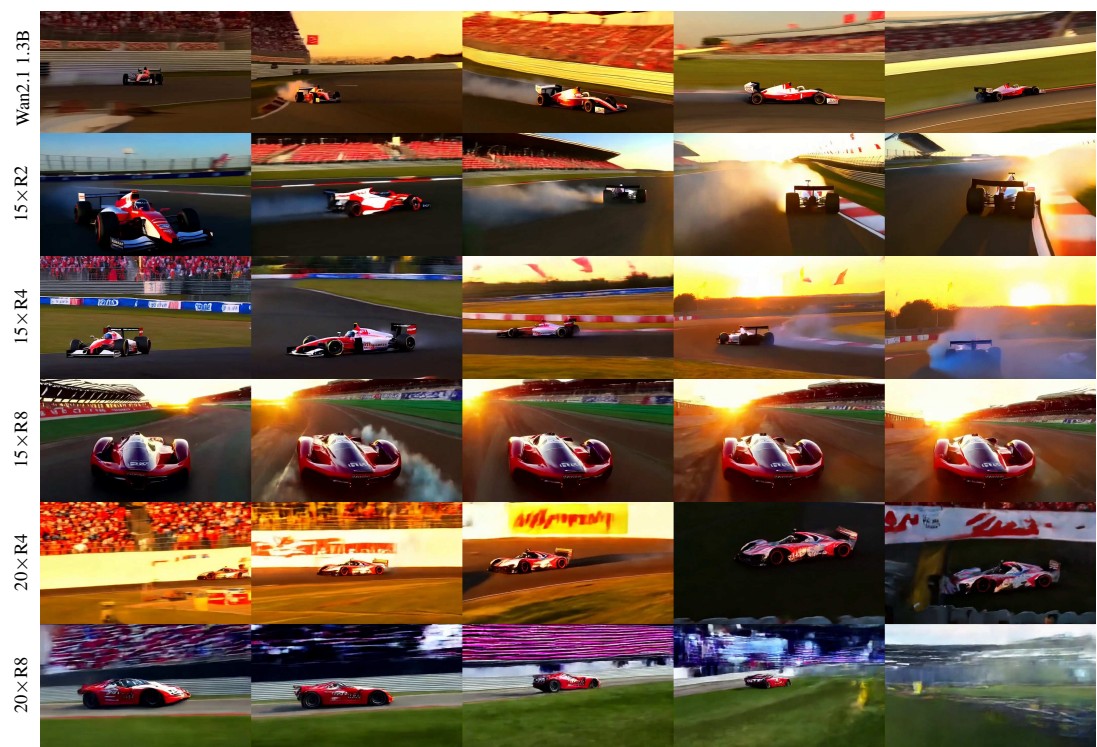

Figure 23: Qualitative videos comparing original Wan2.1 1.3B model to our various hybrid variations for input prompt *raceway*

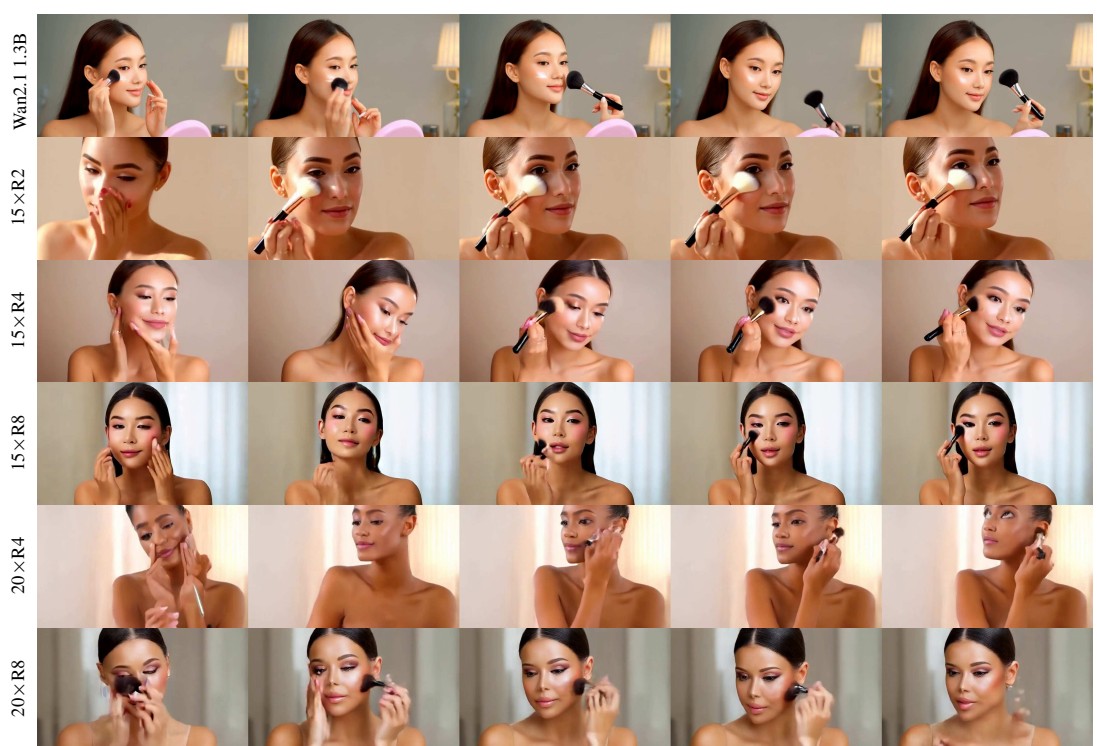

Figure 24: Qualitative videos comparing original Wan2.1 1.3B model to our various hybrid variations for input prompt *this is how I do makeup in the morning.*

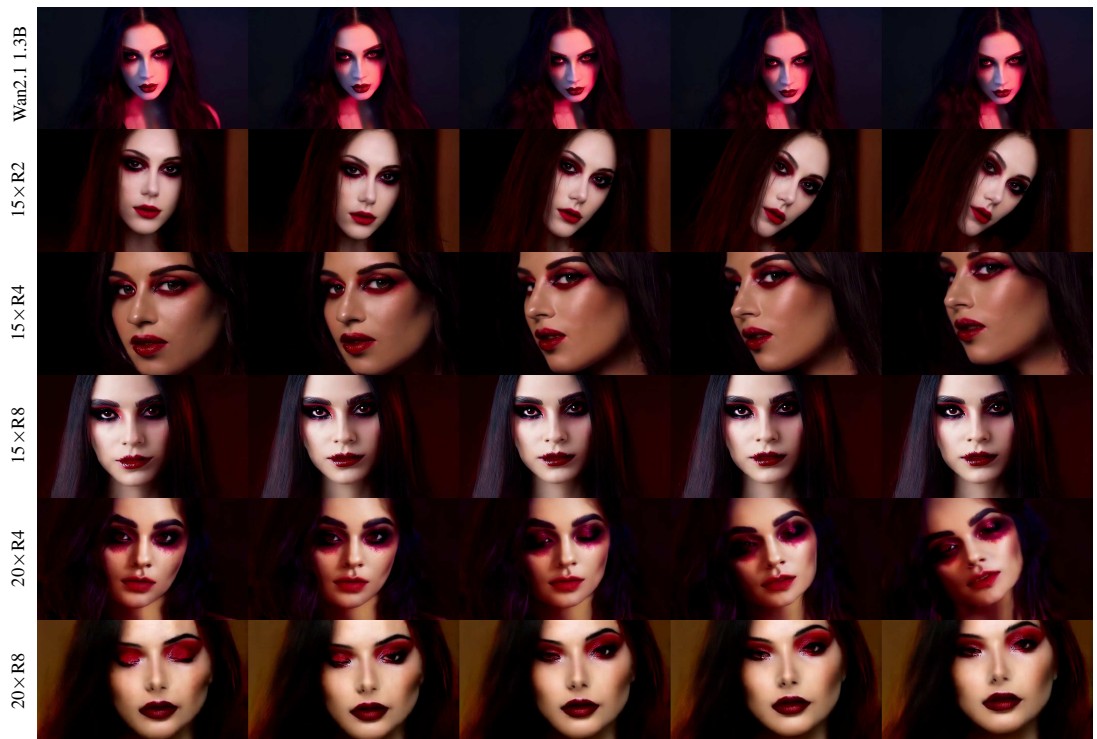

Figure 25: Qualitative videos comparing original Wan2.1 1.3B model to our various hybrid variations for input prompt *Vampire makeup face of beautiful girl, red contact lenses.*

