# OpenReview forum: "Attention Surgery: An Efficient Recipe to Linearize Your Video Diffusion Transformer"
_ICLR.cc/2026/Conference — ICLR 2026 Conference Withdrawn Submission_

### Official Review · Reviewer_AUFE · 2025-10-16

**Soundness:** 1
**Presentation:** 3
**Contribution:** 2
**Rating:** 2
**Confidence:** 4

**Summary:**

This paper proposes a practical framework to convert pretrained video diffusion transformers into efficient sub-quadratic models without retraining from scratch. The method introduces a hybrid attention mechanism that splits tokens into softmax and linear subsets, enabling partial retention of the expressiveness of full attention while reducing computational complexity. It further defines a learnable feature map  using polynomially expanded embeddings to better approximate the exponential kernel of softmax. To retrofit pretrained models efficiently, the authors design a three-stage pipeline—attention distillation, block-rate selection via a multiple-choice knapsack optimization, and lightweight finetuning—requiring less than 0.4k GPU hours. Applied to Wan2.1 (1.3B), the method reduces attention FLOPs by up to 40 % while maintaining video quality on VBench and VBench-2.0 benchmarks

**Strengths:**

1. The paper is very well structured and easy to understand.

2. The hybrid attention formulation and polynomially expanded feature maps  are technically well-motivated.

3. Casting per-block rate selection as a multiple-choice knapsack is elegant and practical.

4. The full conversion pipeline requires less than 0.4k GPU hours, making it appealing for practitioners.

**Weaknesses:**

1. There is no comparison to existing work. The paper does not evaluate against prior efficient attention variants that target video generation, such as Spargeattn, Video Sparse Attention, or Sliding Tile Attention. These methods already demonstrate strong efficiency and quality preservation—achieving over 70% attention reduction and 60% total DiT FLOPs savings. Not only are they absent from the experimental comparisons, but the related work section also fails to discuss them.

Zhang, Jintao, et al. "Spargeattn: Accurate sparse attention accelerating any model inference." ICML 2025.

Zhang, Peiyuan, et al. "Fast video generation with sliding tile attention." ICML 2025.

Zhang, Peiyuan, et al. "Faster Video Diffusion with Trainable Sparse Attention." Neurips 2025.



2. The ablation analysis is insufficient. The hybrid attention design combines softmax and linear components, but it remains unclear whether removing the linear branch would materially affect performance.

3. The main results are generally weak. The paper’s best case achieves only up to 40% reduction in attention cost, whereas previous methods like SpargeAttn and VSA routinely surpass 60–70% while maintaining comparable quality. Given this, it is unclear why a practitioner would adopt Attention Surgery to accelerate video diffusion models—it delivers smaller gains at comparable or higher implementation complexity.

3. There is no discussion of implementation details or system-level impact. The paper presents FLOPs reduction but does not report any wall-clock runtime, memory footprint, or GPU utilization (MFU). It remains unknown whether this hybrid attention can be implemented efficiently with fused kernels such as FlashAttention, or if it requires materializing full attention matrices during distillation—something that would offset the theoretical compute savings.

4. The evaluation scope is limited. All results are reported on the Wan2.1 1.3B model at 480p/320p resolution, with no experiments on larger or higher-resolution models. A natural extension would be to apply the distillation process to Wan 14B at 720p.

**Questions:**

1. What are the actual runtime speedups and MFU compared to Wan2.1 with FlashAttention on GPUs?

2. Can the proposed hybrid attention be implemented as a fused kernel, or does it require materializing attention matrices separately?

---

### Official Review · Reviewer_Pp8E · 2025-10-29

**Soundness:** 3
**Presentation:** 3
**Contribution:** 3
**Rating:** 6
**Confidence:** 3

**Summary:**

This paper focuses on addressing the quadratic complexity growth of self-attention in video models and proposes a hybrid mechanism combining linear and quadratic complexity.

They introduced several interesting improvements, including: (1) a learnable module for queries/keys based on a polynomial mechanism, (2) a novel selection mechanism for "softmax tokens," and (3) an efficient training procedure (attention distillation, lightweight fine-tuning and block-rate optimization strategy), among others.

**Strengths:**

1. What I find interesting about this paper is that it doesn't rigidly adhere to the fundamentalist approach of pure linear attention. Instead, it follows the 80/20 principle by applying quadratic computation to important tokens, achieving a balance between efficiency and quality as explicitly stated in the paper.

2. Their approach to enhancing the capacity of the linear attention module is particularly interesting. The use of polynomial expansion to approximate the large dynamic range of the exponential kernel represents a noteworthy innovation.

3. The experimental results are comprehensive, with well-designed ablation studies (e.g., Tables 3-5) that convincingly demonstrate the importance of each module, making the overall findings highly compelling.

**Weaknesses:**

1. I am somewhat concerned that the performance evaluation is primarily conducted on standard benchmarks like VBench. While useful for controlled comparisons, they may not fully reflect real-world performance on longer, more complex videos. Testing on newer, more challenging benchmarks would strengthen the quality preservation claims.

2. I also have concerns regarding scalability. On one hand, it's unclear whether the approach remains equally applicable as model parameters scale up (e.g., in terms of training/fine-tuning strategies and rate selection mechanisms). On the other hand, there's a lack of empirical validation across different architectures - beyond Wan 1.3B, could it be tested on MMDiT-based models like CogVideo?

**Questions:**

1. I'm curious about whether this method remains equally effective in long-video generation scenarios, such as when applied to Wan models distilled with self-forcing techniques.

2. In Table 5, I observed significantly divergent performance in Dynamic Degree metrics across different distillation losses. What might be the potential explanations for this phenomenon? Could you provide further discussion on these two distillation methods?

---

### Official Review · Reviewer_DgK7 · 2025-10-30

**Soundness:** 2
**Presentation:** 2
**Contribution:** 2
**Rating:** 2
**Confidence:** 5

**Summary:**

This paper proposes Attention Surgery, an efficient framework to linearize or hybridize self-attention in pretrained transformer-based video diffusion models (VDMs) without retraining from scratch. It combines hybrid attention (mixing softmax and linear tokens), lightweight distillation, and cost-aware block-rate optimization to reduce attention computation by up to 40% FLOPs while maintaining generation quality on VBench and VBench-2.0. Validated on Wan2.1 1.3B, the method achieves competitive performance with modest compute costs.

**Strengths:**

1. This method tries to address a critical bottleneck (quadratic self-attention cost) in state-of-the-art VDMs, enabling more efficient long/high-resolution video generation.

2. Lightweight distillation and fine-tuning pipeline avoids prohibitive retraining costs, making it practical for research and industry.
Comprehensive evaluations (quantitative benchmarks + blind user study) robustly validate quality-efficiency trade-offs.

**Weaknesses:**

1. The polynomial feature map for linear attention lacks theoretical justification for its effectiveness in spatiotemporal contexts.

2. Heterogeneous block-rate optimization yields marginal gains (Table 4), raising questions about its added complexity versus benefit.

3. The attention map in Fig. 2 significantly brings the discontinuous memory access problem. Only focusing on the FLOPs is not enough; the authors are encouraged to present the inference efficiency comparison. It's **the most important weakness**.

**Questions:**

Please refer to the weakness part.

---

### Official Review · Reviewer_QQMZ · 2025-11-02

**Soundness:** 3
**Presentation:** 3
**Contribution:** 3
**Rating:** 6
**Confidence:** 2

**Summary:**

This paper proposes Attention Surgery, a recipe to finetune a pre-trained video diffusion transformer into a model with hybrid attention to cut compute while maintaining quality. The key ideas are 1) a token-level hybridization that retains softmax for a uniformly subsampled subset of tokens and applies a learnable linear attention to the rest, 2) lightweight per-block attention distillation (operating on cached teacher trajectories) to match either attention outputs or values, and 3) a heterogeneous block-rate selection formulated as a multiple-choice knapsack problem to assign different hybridization rates per block under a global FLOPs budget. Applied to Wan1.3B, the approach reports up to 40% reduction in attention FLOPs with competitive quality on VBench and comparaoble human preference with the base model, all achieved with a small training budget.

**Strengths:**

- The ability to linearize a pretrained video diffusion transformer into a more efficient, hybrid model has significant practical value.
- The ablation study is very comprehensive. The effect of different distillation loss, block selection, and number of converted blocks, e.t.c., are clearly demonstrated.
- The presentation is very clear and easy to follow.

**Weaknesses:**

- The author mentions that "distilling pre-trained DiT weights into these architectures typically requires extensive training" but I don't see any comparisons with existing softmax-to-linear attention distillation techniques on the same architecture. The paper claims better performance than M4V but it might mostly come from the difference of base model. An apple-to-apple comparison with a prior approach with the same model would strenghten the paper.
- The efficiency gain is only demonstrated in terms of FLOPs, while the wall clock time benefit is unclear.

**Questions:**

Have the author tried to convert all layers (instead of a subset) to hybrid attention? How is the subset 15/20/25 determined? Are there certain layers more suitable for full softmax attention?

---

### Note · Authors · 2025-11-14

**Comment:**

We thank the reviewers for their time and their valuable feedback.

We will work on improving our paper based on the provided input for the next iterations.

**Withdrawal Confirmation:**

I have read and agree with the venue's withdrawal policy on behalf of myself and my co-authors.